# Timing the evolution of phosphorus-cycling enzymes through geological time using phylogenomics

Joanne S. Boden [1] ✉, Juntao Zhong[2], Rika E. Anderson [3] & Eva E. Stüeken [1]

Phosphorus plays a crucial role in controlling biological productivity, but geological estimates of phosphate concentrations in the Precambrian ocean, during life's origin and early evolution, vary over several orders of magnitude. While reduced phosphorus species may have served as alternative substrates to phosphate, their bioavailability on the early Earth remains unknown. Here, we reconstruct the phylogenomic record of life on Earth and find that phosphate transporting genes (*pnas*) evolved in the Paleoarchean (ca. 3.6-3.2 Ga) and are consistent with phosphate concentrations above modern levels (> 3 $\mu$M). The first gene optimized for low phosphate levels (*pstS*; <1 $\mu$M) appeared around the same time or in the Mesoarchean depending on the reconstruction method. Most enzymatic pathways for metabolising reduced phosphorus emerged and expanded across the tree of life later. This includes phosphonate-catabolising CP-lyases, phosphite-oxidising pathways and hypophosphite-oxidising pathways. CP-lyases are particularly abundant in dissolved phosphate concentrations below 0.1 $\mu$M. Our results thus indicate at least local regions of declining phosphate levels through the Archean, possibly linked to phosphate-scavenging Fe(III), which may have limited productivity. However, reduced phosphorus species did not become widely used until after the Paleoproterozoic Great Oxidation Event (2.3 Ga), possibly linked to expansion of the biosphere at that time.

Phosphorus is one of the fundamental building blocks of life. It is an essential constituent of DNA, cell membranes, and ATP, but is quickly depleted by plants and microbes in marine and terrestrial settings worldwide. As a result, restricted phosphorus availability limits primary productivity in large areas of the continents and oceans[1,2].

It is conceivable that Precambrian microbial communities were similarly limited by phosphorus, and if so, that may have constrained the growth and evolution of the biosphere[3–6]. However, geochemical estimates of the Precambrian marine phosphorus reservoir vary by several orders of magnitude, from less than 0.12 $\mu$M to 4,100 $\mu$M in the Archean and between 0.03 and 2.5 $\mu$M in the Proterozoic oceans[7–16].

It has also been proposed that local microscale environments could have achieved up to 100 mM orthophosphate to drive prebiotic chemical pathways[17]. These disparities arise from differences in sampling strategies and assumptions in computational models. Therefore, the extent to which phosphorus was available to microbes in the Archean and Proterozoic remains a topic of debate, especially regarding surface waters with fewer direct constraints.

Modern microbial communities cope with limited phosphorus availability using a rich variety of enzymes to source phosphorus from the environment, including a range of dissolved organic phosphorus (DOP) compounds and inorganic phosphorus species, collectively

[1]School of Earth and Environmental Sciences, University of St. Andrews, Bute Building, Queen's terrace, St. Andrews, Fife, United Kingdom. [2]Department of Medicine, Washington University in St. Louis, St. Louis, MO, USA. [3]Department of Biology, Carleton College, Northfield, MN, USA.
✉ e-mail: j.boden@st-andrews.ac.uk

referred to as total dissolved phosphorus (TDP) (Fig. 1). These include orthophosphate and phosphate esters (where P is in a fully oxidised state; P(V)) as well as more reduced compounds in the form of phosphite (P(III)), phosphonates (likely P(III)) and hypophosphite (P(I))[18–26]. Phosphite and hypophosphite constitute up to 26% of DOP in some habitats[24] and geochemical estimates suggest that phosphite in particular was prominent in the Paleoarchean ocean due to elevated input from meteorites and Earth's more reducing redox state[27,28]. Furthermore, phosphite is a source of phosphorus for multiple microbial species[22], and in some cases, it can serve as an electron donor for chemolithotrophs[25,29,30]. In contrast, phosphonates make up approximately a quarter of all high-molecular weight DOP in the global oceans[31]. They contain carbon chains and characteristically strong carbon-phosphorus (C-P) bonds which are not present in other reduced phosphorus molecules. They can be assimilated as a nutrient resource by a diverse range of bacterial and archaeal taxa across the Tree of Life and in several different habitats[18,19,21–23]. Importantly, genomic studies have found that phosphonate-catabolising pathways catalysed by CP-lyases, phosphonatases, and some dioxygenases are more common in phosphate-depleted waters[20,32] and soils[33], suggesting that they may represent a mechanism for coping with phosphate scarcity. This observation implies that the phylogenetic history of these enzymes can be used as a tool for reconstructing phosphate availability in the past.

Here, we perform phylogenetic analyses to investigate which phosphorus compounds were available for microbial communities over geologic time. More specifically, we reconstruct the evolutionary history of microbial genes involved in utilising phosphate, phosphonates, phosphites, and hypophosphites. To do this, we reconcile a time-calibrated tree of life with phylogenetic trees of phosphorus-utilising genes to estimate when these genes were gained, lost, duplicated, and vertically inherited (see "Methods"). We then infer past phosphate concentrations by comparing presence/absence and abundance patterns for each gene with phosphate concentrations in the modern ocean. Our results allow us to shed new light on the evolution of the biogeochemical phosphorus cycle and provide the first constraints on the availability of reduced phosphorus substrates.

## Results

### Tree of Life

To determine how ancestral microbes used the various phosphorus species considered in this study, we began by reconstructing a tree of life from 865 genomes, chosen to represent the full diversity of life and genomes harbouring *ptxD*. Our results (Fig. 2) are broadly consistent with previous studies. For example, Actinobacteria are sisters to Firmicutes (ultrafast bootstrap support (UBS) = 95) and Proteobacteria probably share a more recent common ancestor with Bacteroidetes (UBS = 66) than with Elusimicrobiota (UBS = 37, Fig. 2) although support values for the latter are low indicating a high degree of uncertainty. A similar relationship has been predicted by Martinez-Gutierrez and Aylward[34] and Coleman, et al.[35], but not Moody, et al.[36].

To estimate when key groups of organisms evolved, we implemented one uncorrelated (UGAM) and two autocorrelated (LN and CIR) Bayesian molecular clock models on alignments of 16 ribosomal proteins that have been used previously to reconstruct the tree of life[30]. The resulting time-calibrated phylogenies made with CIR, LN and UGAM clock models suggest that the last universal common ancestor (LUCA) radiated into Bacteria and Archaea at ca. 4.35 Ga (confidence intervals range from 4.40–4.22 Ga), 4.38 Ga (confidence intervals range from 4.40–4.34 Ga) or 4.37 Ga (confidence intervals range from 4.40–4.29 Ga), respectively. These dates fall shortly after the first evidence for liquid water[37,38], which was applied as an upper constraint for the origin of life.

All three molecular clock models also predict that crown Bacteria emerged at least 200 million years before crown Archaea (Table 1), consistent with previous studies[36]. However, if methanogenesis evolved earlier ≥3.46 Ga instead of ≥2.7 Ga, in line with new geochemical evidence[39–41], crown Bacteria may not have pre-dated Archaea because the estimated origin of crown Archaea is pushed back -0.5 billion years to 3.74 Ga (confidence intervals span 3.83–3.66 Ga) whilst the estimated origin of crown Bacteria remains unaltered at 3.80 Ga with LN clock model (confidence intervals span 3.92–3.68 Ga).

### Distribution of phosphorus-cycling genes in the Tree of Life

We surveyed 865 genomes for 13 genes involved in the utilisation and production of reduced phosphorus molecules and phosphate.

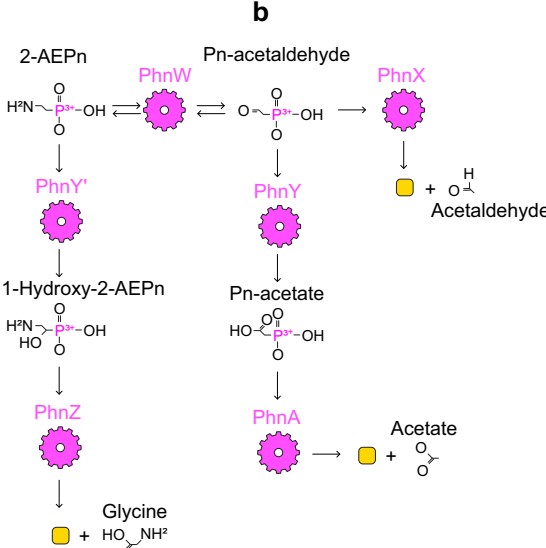

**Fig. 1 | Enzymatic pathways for importing and catabolising reduced phosphorus molecules and phosphate.** An overview of the metabolisms researched in this project (**a**) and a detailed summary of substrate-specific phosphonate degradation pathways (**b**). Coloured squares indicate hypophosphite (grey), phosphite (blue), phosphonate (pink) and phosphate (yellow) molecules, whereas other shapes of the same colour indicate the enzymes responsible for catalysing their import and catabolism.

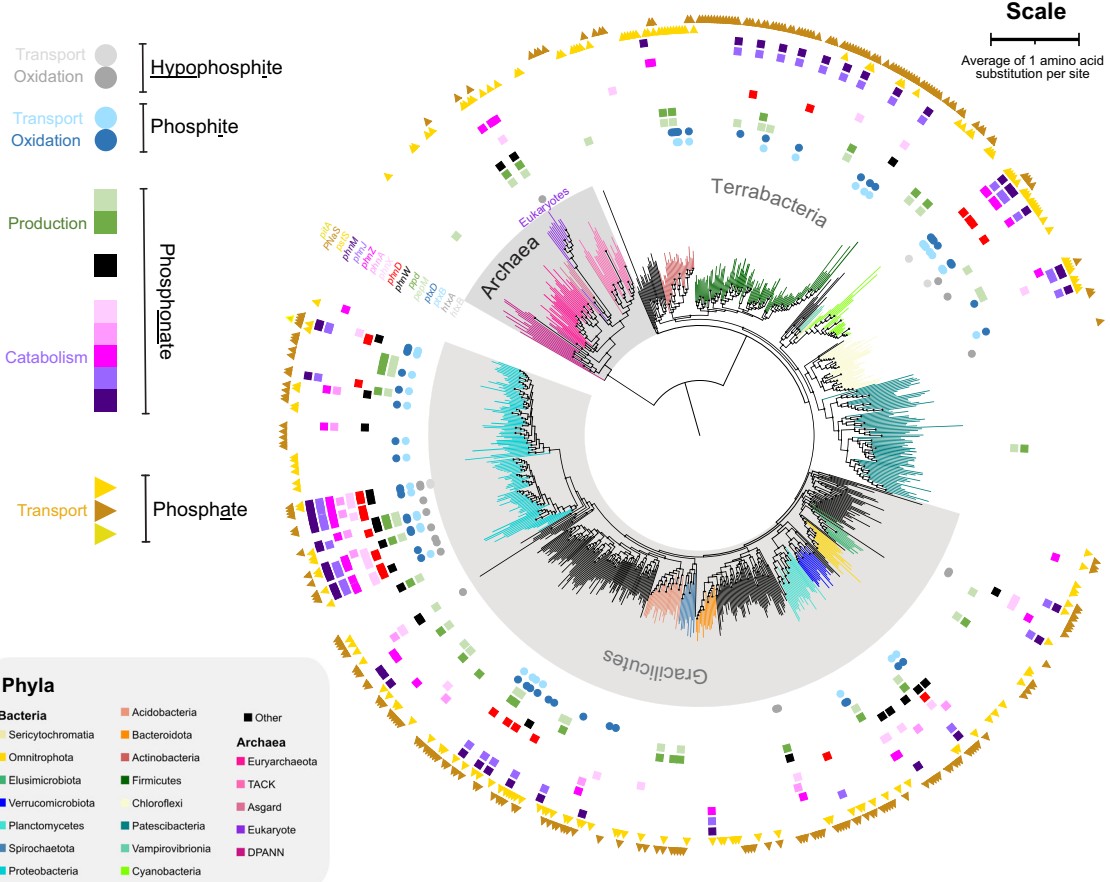

**Fig. 2 | Distribution of phosphorus-cycling genes in the Tree of Life.** Coloured shapes outside branch tips represent homologs of enzymes involved in hypophosphite, phosphite (circles), phosphonate (squares) and phosphate (triangles) utilisation and production. Branches are coloured according to their phylum. The evolutionary tree was reconstructed from 16 ribosomal proteins using maximum likelihood methodology. The scale bar represents an average of one amino acid substitution per site. The complete tree with support values is available in Newick format in our open science framework repository, https://osf.io/vt5rw/?view_only= b13a53f4d87c44d1a82a18b176523c5b.

The results reveal a broad distribution of genes encoding phosphate sodium symporters (namely *pnas*) and active phosphate transport mechanisms (indicated by *pstS*) throughout 257 and 174 strains of our dataset respectively (Fig. 2). These encompass all major bacterial and archaeal phyla except DPANN and Patescibacteria (formerly CPR, Fig. 2). In contrast, genes for low-affinity phosphate import of orthophosphate ions bound to divalent metal ions, such as $Mg^{2+}$, $Ca^{2+}$, $Co^{2+}$, $Mn^{2+}$ and $Zn^{2+}$ via *pitA* and *pitH* are absent or rare respectively. Only one of the 865 genomes harbours a *pitH* homolog (Fig. 2).

Reduced phosphorous metabolisms are present in a variety of bacteria, but most archaea cannot metabolise phosphonates, phosphite or hypophosphite (Fig. 2). Of these, bacteria from the Proteobacterial superphylum host a particularly broad variety of phosphonate-cycling genes, as has been documented in previous studies[20,21]. We also find that phosphonate production is rarer than its catabolism, (consistent with previous work[18]), because only 30 genomes harbour *pepM* and *ppd* to produce phosphonates from phosphate esters, whereas 94 genomes have at least one of *phnX*, *phnA*, *phnZ*, *phnJ* or *phnM* to catabolise phosphonates. Of all the phosphonate catabolism genes, *phnJ* and *phnM* encoding the broad-specificity CP-lyases are most common because they co-occur in 44 genomes compared to the substrate-specific phosphonate catabolism genes, *phnX*, *phnW*, *phnA* and *phnZ* which are each present in just 28, 25, 12 and 40 genomes respectively (Supplementary Table 1). The distribution of these genomes in the Tree of Life (Fig. 2) reveals a broader taxonomic distribution of phosphonate-producing compared to phosphonate-catabolising organisms, which is consistent with previous work[20].

Hypophosphite is more reduced than phosphonates and phosphite. As a result, only 4 genomes have *htxB* encoding the binding protein required for active hypophosphite import from the environment. Two of these genomes also have *htxA* for oxidising hypophosphite to phosphite, but a further 15 genomes have *htxA* without the binding protein for hypophosphite import (Fig. 2). these *htxA* genes may encode enzymes that oxidise formate instead, as biochemical analyses have found comparable rates of activity on these small organic molecules[42].

## Timing of gene events

To estimate when these phosphorus-cycling genes emerged and radiated across the tree of life, our molecular clocks were reconciled with Bayesian evolutionary trees of each phosphorus-cycling gene to ascertain the timing of speciation, duplication, loss, and horizontal gene transfer events. The most ancient event for a given gene represents the first phylogenetic evidence of that gene's presence, whereas a clustering of events in a particular time interval represents widespread use of the metabolism across diverse lineages. Any lineages which experienced gene events but subsequently became extinct do not leave signatures in the genomic record, so in these cases the origin of each gene could be earlier than we report. Our estimates represent lower bounds.

**Table 1 | Molecular clock estimates timing of the emergence of crown bacteria and archaea under three different clock models**

| Clock Model | Bacteria / Ga | Archaea / Ga |
|---|---|---|
| CIR | 3.47 (3.61–3.31) | 3.26 (3.45–3.07) |
| LN | 3.80 (3.91–3.67) | 3.26 (3.45–3.08) |
| UGAM | 3.90 (4.28–3.57) | 3.32 (3.51–3.13) |

Confidence intervals are provided in brackets.

Estimates made from different clock models vary, with UGAM predicting generally younger events than LN and CIR (Supplementary Figs. 1–14) but all three suggest that microbial communities began importing phosphate before they evolved methods of assimilating reduced phosphorus molecules in the form of phosphonate, phosphite and hypophosphite. For example, the earliest evidence of the phosphate-sodium symporter *pnas* appears in the Paleoarchean, whilst the earliest evidence for phosphite import and oxidation emerges in the Neoarchean (Fig. 3). For the rest of this paper, we focus on estimates from the CIR clock model because it has been shown to outperform other clock models in terms of model fit[43] and consistency between lineages participating in the same HGT events[40].

Using this clock model, we conducted further tests to see how ecceTERA's reconciliation parameters[44] impact the estimated histories of phosphorus-cycling enzymes. Duplication, loss, and transfer events involving each phosphorus-cycling gene are predicted based on three things: The shape of the gene tree and molecular clock, the estimated ages of lineages participating in the same transfer events, and the expected cost of transfer in relation to duplication, loss, or speciation of the gene. The latter is a proxy for the general difficulty of duplication in comparison to loss and transfer, so costs remain the same regardless of which genes or species are being assessed. Default costs assume that transfer is more expensive than duplication, loss and speciation in a ratio of 3:2:1:0 because these result in the least genome size variation between parent and daughter lineages[45]. However, genome size variation is difficult to predict on multi-million-year timescales (Supplementary Discussion), so we conducted reconciliations with different HGT costs (namely 2,4, and 6) to represent scenarios with more genome size variation[45]. We found that different costs of horizontal gene transfer (HGT) result in older events for *pstS*, *phnM* and *phnJ* (Supplementary Figs. 15 and 16), but as stated above, microbial communities always begin importing phosphate before assimilating reduced phosphorus molecules, and the earliest evidence for phosphate and phosphite import are in the Paleoarchean and Neoarchean respectively (Supplementary Figs. 15 and 16). For the rest of the manuscript, we report on the findings of all costs tested.

Microorganisms began catabolising phosphonates with the evolution of *phnX* in the Mesoarchean, but the ability remained confined to a small subset of lineages in comparison to phosphate import mechanisms (Fig. 3, Supplementary Figs. 15 and 16). The gene *phnX* encodes phosphonatase, which acts on phosphonoacetaldehyde to produce phosphate and small organic molecules for downstream metabolisms within the cell (Fig. 1). As time progressed, the number of potential phosphonate molecules that could be used as a source of phosphorus increased with the emergence of *phnW*, encoding 2-aminoethylphosphonate-pyruvate transaminase, which can act in concert with *phnX* to release phosphate from 2-aminoethylphosphonate (2-AEP) (Figs. 3, and 1). This enzyme is estimated to have emerged in either the Mesoarchaean (HGT cost of 6), Neoarchean (HGT cost of 3 or 4) or Paleoproterozoic (HGT cost 2) depending on the event cost (Supplementary Figs. 15 and 16). The genomic repertoire for catabolizing phosphonates also increased in the Paleoproterozoic with the spread of the genes *phnJ* and *phnM*, encoding key components of broad-specificity-CP-lyases, which

emerged in the Mesoarchean, Neoarchean or start of the Paleoproterozoic depending on event cost (Supplementary Figs. 15 and 16). These genes would have enabled micro-organisms to liberate phosphate and glycine from a post-cursor of 2-AEP and to catabolise a wide variety of different phosphonates molecules (Fig. 3).

Biological methods of producing these phosphonates began to emerge during the Great Oxidation Event (GOE), with the evolution of *pepM* for transforming the relatively oxidised phosphate ester phosphoenolpyruvate (P in +5 oxidation state) to the more reduced phosphonate phosphonopyruvate (P in +3 oxidation state) and vice versa. This reaction is reversible, so another enzyme is required to force the reaction in the direction of phosphonate production; *ppd* (otherwise known as *aepY*) encodes phosphonopyruvate decarboxylase, which is one of the most common methods of doing this[19]. Here, we find phylogenomic evidence to suggest that *ppd* emerged less than 300 million years after *pepM* (Fig. 3, Supplementary Figs. 15 and 16).

The earliest evidence for microbial utilisation of phosphite and hypophosphite appeared in the Neoarchean (or late Mesoarchean if the HGT cost is very high at 6) with the advent of *ptxB* for importing phosphite and *htxA* for oxidising hypophosphite and formate (Fig. 3, Supplementary Figs. 15 and 16)[42]. It has previously been hypothesised that dissimilatory phosphite oxidation (DPO) emerged in the most recent common ancestor of Firmicutes and Deltaproteobacteria[25], which emerged ~3.2 Ga[46] or ~2.9 Ga[47] depending on which molecular clocks are used to infer their age. Here, we predict that this ancestor arose ~3.45 Ga–~3.85 Ga depending on which clock model is used (Supplementary Table 2), but our gene-tree-species-tree-reconciliations do not find evidence of *ptxD* in this ancestor. Instead, the earliest event indicative of the presence of *ptxD* is speciation which occurred within the Firmicutes 2.30 Ga (confidence intervals span 2.56–2.22 Ga, Fig. 3). This could range back to 2.83 Ga if HGT was more difficult (Supplementary Fig. 15c), but it places an upper limit on the origin of DPO in the Meso-Neoarchean boundary or more recently unless *ptxD* was present in an earlier lineage which subsequently went extinct.

## Discussion
### Reconstructing phosphate availability through time

Our results allow us to place new constraints on phosphate concentrations in Precambrian environments. Many reduced-phosphorus cycling genes are only present in microbial communities when more easily accessible forms of phosphorus, such as orthophosphate, are not available[20,22,32,33]. We performed regression analyses of data from the *Tara* Oceans Project[48,49] and found significant correlations between the abundance of four genes in our dataset and phosphate concentrations in the surface ocean (Fig. 4). Gene abundances of *pstS*, *phnM* and *phnJ* reduce with increasing phosphate concentrations (coefficients are −0.79, −0.29 and −0.32 respectively), whereas the abundance of *nptA* from the *pnas* symporter family decreases (coefficient = $4.78 \times 10^{-5}$, Fig. 4). For the remaining nine genes, the relevant data are not available. These trends imply that genes encoding CP-lyases and high-affinity phosphate transporters are present in low phosphate environments and relatively rare in regions with phosphate concentrations approaching 3.29 µM (Fig. 4). Similar negative correlations between marine phosphate concentrations and phosphonate-cycling genes, including CP-lyases, phosphonatase (encoded by *phnX*) and Fe(II)-dependent phosphonohydrolases (encoded by *phnZ*) have been documented previously in global surface oceans[20,32] and terrestrial soils[33]. In contrast, *nptA* which is a member of the low affinity PNaS symporter family, is more abundant in regions with high phosphate concentrations (>1 µM) (Fig. 4).

These correlations, combined with our estimates on the origin of phosphate transporters and broad-specificity CP-lyases can be used to provide estimates on phosphate availability in the Archean and Paleoproterozoic. Geochemical studies have proposed dissolved phosphate concentrations ranging from less than 0.004[8–11] up to

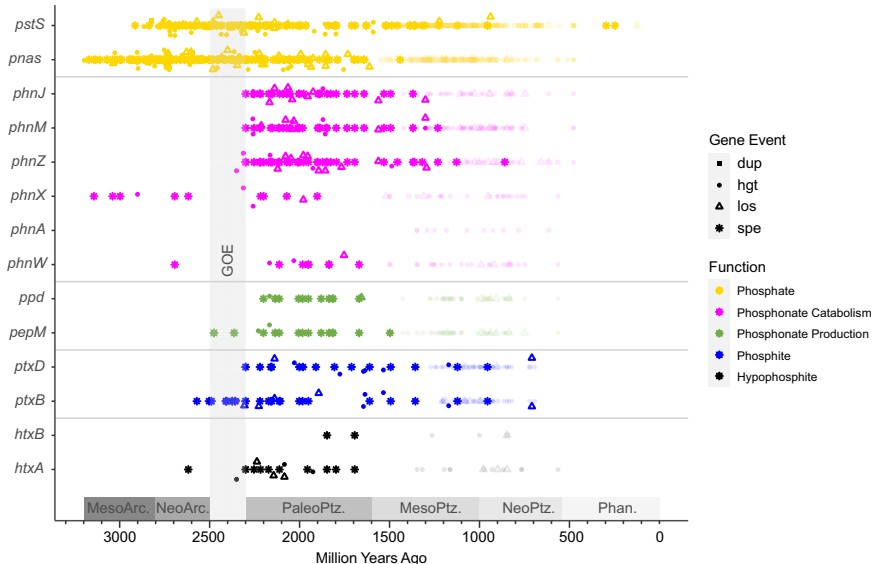

**Fig. 3 | Estimated origin of phosphorus-cycling genes in the Tree of Life assuming relative event costs of 3:2:1:0 for transfer, duplication, loss, and speciation respectively.** Results of other event costs are available in the supplementary information Figs. 15 and 16. Coloured shapes represent speciations (asterisks), horizontal gene transfers (filled circles), duplications (filled squares) and losses (empty triangles) of genes underlying phosphate utilisation (yellow), phosphonate usage (pink), phosphonate production (green), phosphite usage (blue) and hypophosphite usage (black). Each transfer, duplication and loss is estimated to have occurred on either an internal (bright colours) or terminal (faded colours) branch of our time-calibrated tree-of-life, made using the Cox-Ingersoll-Ross clock model. Our methods do not predict whether transfers, duplications and losses happened at the start or end of the branch, so we summarise them by plotting the midpoints. Speciations are plotted on the node where they occur. For a full representation of branch lengths and uncertainties, see supplementary information Figs. 1–14.

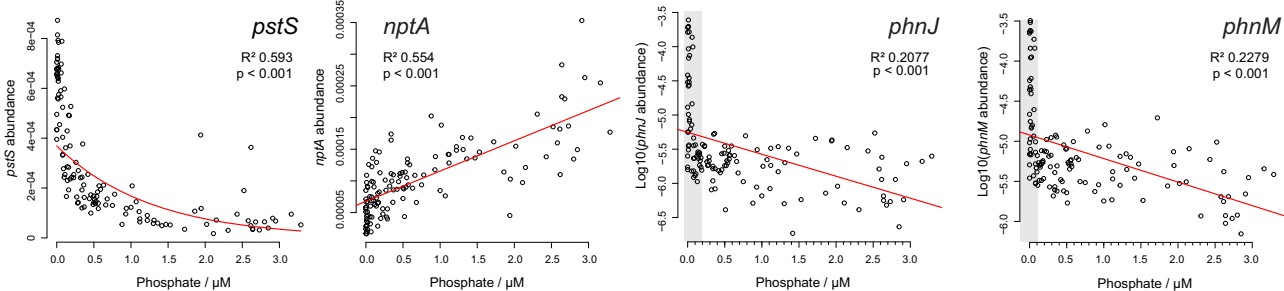

**Fig. 4 | Regression analyses of phosphorus-cycling genes in regions of the global ocean with varying phosphate concentrations.** The abundance of reads encoding part of *pstS*, *nptA*, *phnJ* and *phnM* are plotted on the *y*-axis against micromolar phosphate concentrations on the *x*-axis, for all 139 samples from the Ocean Microbial Reference Catalogue v2 (OM-RGC.v2) of the *Tara* Oceans Project[44,45]. Regression lines are shown in red and represent the results of two-sided tests. The resulting p values are $<2.2 \times 10^{-16}$ for *pstS* and *nptA*, $4.5 \times 10^{-8}$ for *phnJ* and $8.2 \times 10^{-9}$ for *phnM* with 129 degrees of freedom. Grey areas indicate low phosphate concentrations where the abundances of CP-lyase genes are particularly high.

4100 μM[7] in the Archean Ocean (Supplementary Fig. 17), which is up to ~1000 times higher than the highest phosphate concentration in the modern ocean (0–3.29 μM were measured in the *Tara* dataset). Given the positive linear correlation of the sodium-phosphate symporter, *nptA*, with phosphate availability (Fig. 4), it is likely that the organisms which first acquired these phosphate-sodium symporters were exposed to higher than modern phosphate concentrations during the Paleoarchean (Fig. 3, Supplementary Figs. 15 and 16) when the first evidence of enzymes from this family appears. Whether they were present in organisms from terrestrial, marine or subsurface environments is unclear, but the phosphate concentrations they indicate are consistent with the global geochemical estimates of Ingalls, et al.[13], Brady, et al.[7] and Rasmussen, et al.[12].

When *pstS* evolved, which could either have been toward the end of the Mesoarchean (Fig. 3) or with PNaS in the Paleoarchean (Supplementary Fig. 15), areas of the planet must have experienced low phosphate concentrations to drive the emergence of more metabolically expensive methods of obtaining phosphorus using high-affinity phosphate importers (Fig. 3). Extrapolations of our exponential regressions suggest that at 1000 μM phosphate, *pstS* would be extremely scarce (Fig. 4). At 25 μM phosphate, *pstS* abundance would also be very low, which would probably not lead to a strong selective pressure driving the evolution of the gene (Fig. 4). As a result, we consider it more likely that the Paleoarchean or late Mesoarchean organisms (Fig. 3) in which *pstS* first evolved would have occupied areas with low phosphate concentrations of 1 μM or less, in line with geochemical estimates by several studies[8,9,11,15,16,50]. By extension, similar areas with low phosphate must have persisted throughout the Neoarchean, Proterozoic and Precambrian to drive the speciation, duplication, and gene transfer events of *pstS* that are predicted in our molecular clocks (Fig. 3).

The relationships of *phnJ* and *phnM* with phosphate provide further insight into more recent Paleoproterozoic or Mesoarchean environments because they emerge a few hundred million years after both phosphate uptake mechanisms in all reconstructions (Supplementary Figs. 15 and 16). These genes, *phnJ* and *phnM*, are key

components of CP-lyases which enable micro-organisms to use phosphonates as a source of phosphorus and small organic molecules. In the surface ocean, microorganisms seem to use phosphonates as a source of phosphorus because *phnJ* and *phnM* abundances exhibit a stark drop-off above 0.1 μM phosphate. Above this threshold the highest abundances of *phnM* and *phnJ* drop by an order of magnitude and median values drop by a factor of 4–5 (Fig. 4). In the deeper ocean (from 200 to 1,000 m depth), phosphate is more readily available (mean concentration of 2.1 μM with a range from 0.18 to 3.29 μM compared to 0.37 μM with a range from 0 to 2.31 μM at the surface[49]), so there are fewer *phnJ* and *phnM* genes in general, and they are not more abundant when there is less phosphate[20]. As a result, phosphonates may be catabolised primarily as a source of carbon in phosphate-replete environments, and primarily as a source of phosphorus in phosphate-depleted environments[20]. This raises the question of whether the availability of other nutrients such as carbon could influence the evolution of *phnJ* and *phnM*. As there are fewer genes encoding CP-lyases in the phosphate-replete environments of the mesopelagic compared to the phosphate-depleted environments of the surface ocean[20], environments with very low phosphate concentrations (below 0.1 μM) are likely to have driven the evolution of broad-specificity phosphonate catabolism in the Paleoproterozoic or Mesoarchean, depending on the difficulty of HGT. This is within the range predicted by some geological records of Paleoproterozoic deep ocean habitats (e.g., refs. 10,16 and ref. 9), but lower than others (e.g., ref.[11]).

To understand whether our estimates of phosphate availability are relevant for terrestrial or marine environments of the Paleoproterozoic and Archean, we need to know where the ancestral organisms which harboured *pnas*, *pstS*, *phnJ* and *phnM* were living. Some insight can be gained from our inferred gene events. Different ancestral lineages are predicted to have inherited phosphorus-cycling genes depending on which clock model was applied (Supplementary Figs. 18 and 19), but all reconciliations, regardless of the clock model, predict that Archean gene events involving *pstS* and *pnas* were numerous and occurred in a range of bacterial phyla, including Bacteroidota, Actinobacteria, Chloroflexi, Omnitrophota, Acidobacteriota, Cyanobacteria and Firmicutes (Supplementary Fig. 18). In contrast, more recent events involving *phnJ* and *phnM* were rarer and occurred in Verrucomicrobia, Firmicutes, Bacteroidota, Cyanobacteria and Proteobacteria (Supplementary Fig. 19). Members of these phyla today occupy many different habitats, including the terrestrial subsurface[51] and marine water column[52]. Therefore, it is possible that a large range of local geographical areas exhibited similar or lower than modern phosphate concentrations in the Archean and Paleoproterozoic.

Future research indicating the concentrations of phosphonate, phosphite and hypophosphite in relation to the abundances of genes for their uptake and/or oxidation and catabolism is required to link genomic data to the concentrations of reduced phosphorus molecules in the environment. As it stands, we cannot comment on the concentrations of these reduced phosphorus compounds other than concluding that they must have been present and beneficial to organisms when enzymes that act upon them evolved.

### Paleoenvironmental and ecological implications

By tracing the evolutionary trajectories of enzymes for importing and utilising a range of phosphorus species, we have provided insight into the microbial usage of phosphorus compounds throughout the Precambrian. Results predicted using default event costs, which estimate a late Paleoarchean emergence of phosphate sodium symporters (encoded by *pnas)* that are currently abundant in phosphate-replete environments, followed by phosphonatase and a high-affinity phosphate transport binding protein (encoded by *pstS*) that are most abundant in phosphate-depleted environments (Figs. 3 and 4)[20], suggest that at least locally phosphate-enriched habitats were present in

the early Mesoarchean and regions limited by phosphate spread later toward the end of the Archean (Fig. 5). We speculate that this drop, if global, may be linked to increased deposition of banded iron formations (BIF) in the Neoarchean compared to the Mesoarchean[53]. If primary BIFs were composed of ferric (Fe(III)) iron oxides, which have a high affinity for phosphate adsorption, they may have scavenged phosphate from the water column[15,16]. In contrast, phosphate would have been relatively more soluble in the presence of Fe(II) only, possibly allowing phosphate concentrations of hundreds to thousands of μM[7]. However, it is important to note that this delay between low and high-affinity phosphate uptake mechanisms and the associated drop in phosphate levels is not estimated in other reconciliations with different event costs (Supplementary Fig. 15). We also emphasise the origin of BIFs is debated and other mechanisms, such as a purely biological phosphate drawdown, may have dominated.

A drop in phosphate from the early to late Mesoarchean, if true, could have limited biological productivity and $O_2$ production by Cyanobacteria, which perhaps delayed the rise of $pO_2$ in Earth's atmosphere, i.e., the GOE, to 2.45 Ga, long after the evolutionary origin of oxygenic photosynthesis. Indeed, our limited analyses of 22 cyanobacterial genomes included in the molecular clocks and reconciliations with default event costs predict that Cyanobacteria were genomically capable of using phosphate sodium symporters in the early Neoarchean, and only acquired *pstS* for active phosphate import toward the end of the Neoarchean (Supplementary Fig. 19).

Whilst multiple microbial species from different phyla were genomically capable of importing phosphate in the Archean, fewer species could use reduced phosphorus compounds (as indicated by the relative scarcity of events in Fig. 3 and Supplementary Figs. 15 and 16). This appears to have changed at the Neoarchean to Paleoproterozoic boundary, when phylogenetic evidence points toward more widespread dissemination of genomic mechanisms for assimilating phosphite (via *ptxD*) and a broad variety of phosphonate molecules (via *phnZ* and potentially *phnJ* and *phnM*). This post-GOE expansion in the phosphorus pool available to microbes may be a consequence of the growing biosphere[50,54]. Some biogeochemical models suggest that primary productivity was limited by the availability of electron donors until the evolution of oxygenic photosynthesis, which became widespread at the start of the GOE[54]. Similarly, models of ancient carbon cycles suggest that marine net primary production increased by two orders of magnitude from the Neoarchean to Paleoproterozoic[50]. As microbial communities grew, more and more organisms would have competed for the available phosphorus, perhaps increasing the selective pressures driving the evolution of genomic mechanisms for utilising alternative phosphorus species.

The evolution of biological phosphonate production with the emergence of *ppd* in the Paleoproterozoic may also have been a result of more crowded microbial habitats as phosphonates can act as competitive inhibitors of phosphate and carboxylic acid functional groups in metabolic pathways, interrupting the growth of other microorganisms[18]. They are more resistant to degradation than phosphate esters, owing to their stable C-P bonds, which has led to the hypothesis that marine microbes produce them to protect against viral infection and grazing[18]. We note that biological phosphonate production appears to have emerged after phosphonate utilisation in our data, which may indicate an extinct phosphonate-generating pathway in the early Archean that was not sufficiently widespread to be preserved in the genomic record. Alternatively, it is possible that minor phosphonates were generated abiotically in metamorphic or hydrothermal settings in the presence of phosphate and organic matter, although experimental evidence for this is so far lacking. In any case, our results unambiguously show the Paleoproterozoic radiation of biological phosphonate catabolism pathways after their Mesoarchean to Paleoproterozoic origin, and the observation that this signal is

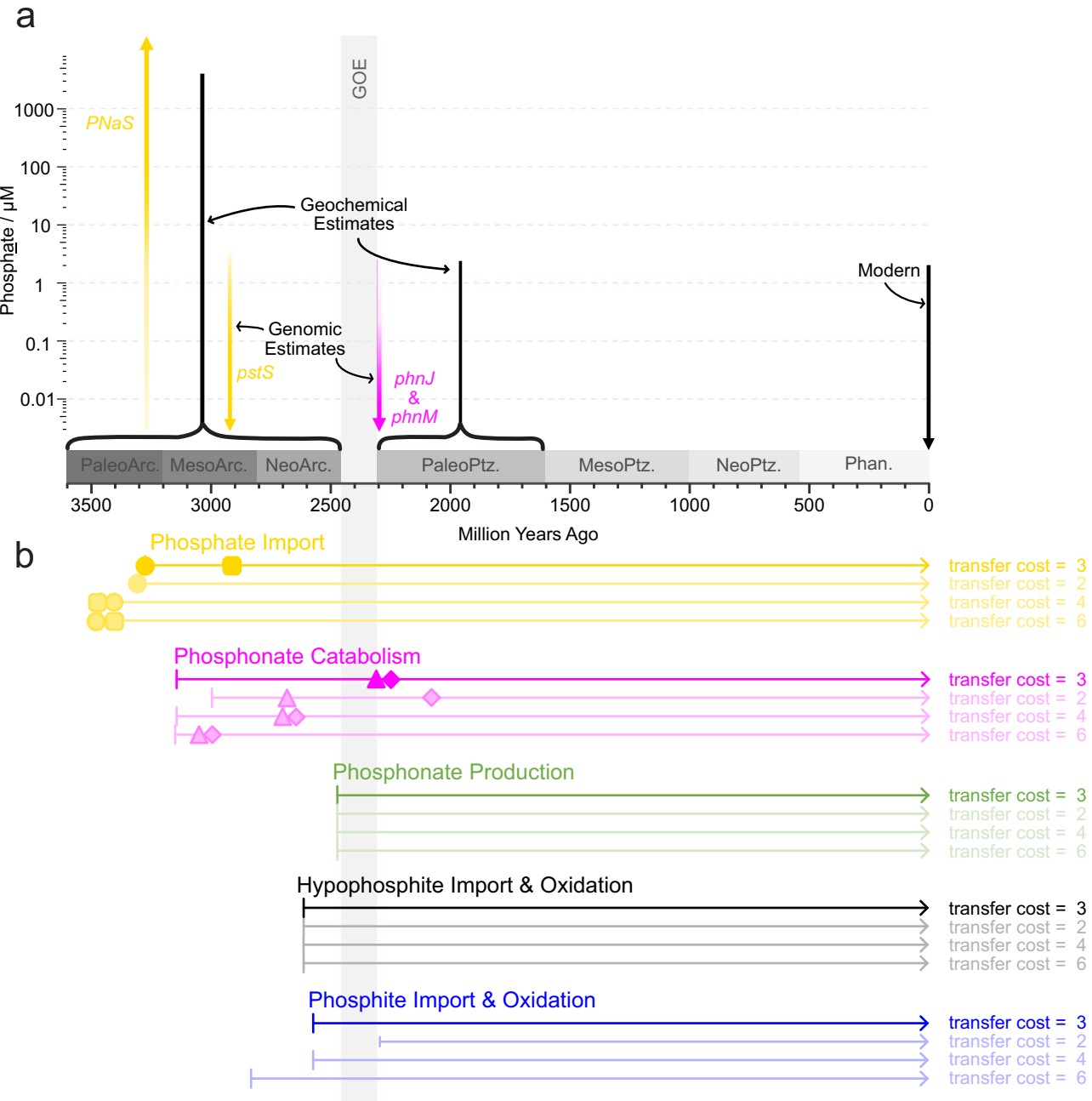

**Fig. 5 | Estimates on the availability of orthophosphate based on genomic records.** Geochemical estimates of marine phosphate concentrations (vertical black bars) for the Archean and Proterozoic are indicated alongside phosphate concentrations most likely to have driven the evolution of the phosphate transport enzymes PstS and PNaS (vertical yellow arrows) and the phosphonate-metabolising enzymes PhnJ and PhnM (pink bars) according to divergence time estimates made with the CIR clock model and event costs of 3:2:1:0 for transfer, duplication, loss and speciation (**a**). Coloured horizontal lines in panel (**b**) represent the time since enzymes involved in the microbial utilisation of phosphate (yellow), hypophosphite (black), phosphite (blue) and phosphonate (pink) are estimated to have existed when using different reconciliation parameters. Green horizontal lines represent the time since enzymes which produce phosphonates have existed. Each line spans the period since the first gene encoding an enzyme in the metabolism evolved, with the earliest estimates for *pnas* (yellow circle), *pstS* (yellow square), *phnJ* (pink triangle) and *phnM* (pink diamond) highlighted with coloured shapes. GOE, Great Oxygenation Event.

preserved in modern organisms speaks to the significance of this metabolic pathway at that time.

In conclusion, our phylogenetic analyses provide insight into the bioavailability of phosphorus molecules and the concentrations of phosphate throughout the Archean and Proterozoic. We provide the first estimates on the origin of alternative phosphorus species and use an independent approach to estimate Precambrian phosphate concentrations around microbial communities. Together, these suggest that phosphite and hypophosphite were not widely used by the ancestors of present-day biota for most of the Archean. Phosphonate use dates back to the Mesoarchean, but like phosphite and hypophosphite they appear to have become more widely used after the GOE, possibly linked to biological diversification and increased competition. Lastly, our results advance ongoing debates about the Precambrian phosphate reservoir and help reconcile previously conflicting geochemical estimates. Phosphate-replete conditions inferred from

solubility experiments[7] might be consistent with genomic data for the early emergence of low-affinity phosphate uptake mechanisms in the Paleoarchaean, whereas phosphate-depleted conditions proposed by others[8,9,11,15,16,50] took over toward the end of the Archean into the Paleoproterozoic, as supported by the emergence of multiple phosphonate catabolism pathways and potentially high-affinity phosphate transporters at that time. Phosphate limitation may therefore have been a significant controlling factor in the timing of the GOE.

## Methods

### Species tree reconstruction

Our evolutionary tree of life was constructed from information present in the ribosomal proteins of 865 genomes (Supplementary Dataset 1), which were chosen to represent the full diversity of bacteria and archaea using methods described in Mateos, et al. [55]. To achieve this, we included one representative from each order of the tree of life using information from the Genome Taxonomy Database (GTDB) release 95[56,57]. Where possible, a representative with the gene *ptxD* was included to better capture the diversity of phosphite-oxidising organisms. We also include six cyanobacteria, six plastids and four algae to enable us to calibrate the molecular clock using microfossils spanning a range of geological time intervals. From these genomes, a species tree was reconstructed using 16 ribosomal proteins (namely L18, L3, L5, S8, L4, S3_C, L6, L2, L15e, S19, S17, L22, S10, L24, L16 and L14) that have been used previously to infer the tree of life[58-60]. Different congregations of 14 to 40 proteins[34,36,47,58,61-64], sometimes 381[65] or 400[66], have also been used to reconstruct the tree of life, but most include ribosomal proteins (Supplementary Table 3) because they are ubiquitous and resistant to horizontal gene transfer[67]. Therefore, homologues of our 16 ribosomal proteins were obtained from each genome with HMMER3 v3.3.2[68] (95 % of the genomes contained homologues of eight or more of these genes), then aligned and trimmed with Muscle v5.1[69] and TrimAl v1.4.rev15[70], all implemented in GToTree v1.6.34[71].

The resulting alignments were visualised in AliView v.1.28[72] and used to reconstruct a species tree using the maximum likelihood methodology implemented in IQTREE v2.0.3[73]. Partitioned analyses[74] were applied to allow each protein to evolve under appropriate substitution models. To find these substitution models, ModelFinder was implemented with default options (no mixture models) and 'MERGE'[75]. The best was found to be a single partition containing all proteins, so the LG substitution rate matrix was applied to all 16 ribosomal protein alignments, with 10 categories of the FreeRate model to estimate substitution rates and their variations at different sites. Further variations in substitution rates through time (as described in Lopez et al. [76]) were modelled by applying the -Q option to account for heterotachy.

Species trees made using different methods sometimes estimate different relationships between phyla (e.g., as demonstrated her[64]), so we incorporated information from previous studies which thoroughly investigated macroevolutionary relationships between large clades of bacteria and archaea (including[34,35,47,65,77,78]) by applying a topological constraint on the relationships between Gracilicutes, Terrabacteria, Patescibacteria/CPR and Euryarchaeota (including Halobacteriota, Thermoplasmatota, Methanobacteriota, Hydrothermarchaeota and Hadarchaeota) in our tree (Supplementary Fig. 20). For all analyses, branch supports were estimated with 1000 replicates of ultrafast bootstrap approximations[79] and the SH-aLRT test[80]. Three independent replicates were completed, but only one consensus tree with the best (a.k.a. closest to zero) log-likelihood value was retained for further analyses. The resulting phylogenies were visualised in TreeViewer v2.1.0[81], FigTree v.1.4.4[82] and iTOL v6.8[83].

### Divergence time estimation

Our species tree was anchored to a geological timeline by implementing a relaxed Bayesian molecular clock in Phylobayes v.4.1[84].

Topology was fixed to match our species tree, and ages were estimated based on predicted mutation rates for the 16 ribosomal proteins described above (2290 aligned amino acid positions). Substitutions were modelled using a uniform exchangeability matrix and an empirical profile mixture model with 20 profiles (specified using -catfix C20)[85]. Divergence times were estimated under one of three models to reflect differing views on the inheritance of substitution rates between mother and daughter lineages: These were the autocorrelated lognormal (LN) and Cox-Ingersoll-Ross (CIR) models where substitution rates are inherited from parent lineages[43,86], and the uncorrelated gamma multipliers (UGAM) model where substitution rates are chosen independent of ancestry[87].

The first diversification in our clock, representing the first radiation of LUCA into bacteria and archaea, was set by applying a uniform distribution spanning 4.4–3.5 Ga to the root node. This is based on the earliest evidence for liquid water on the planet's surface[37,38] and some of the earliest generally accepted microfossils[88].

We also implemented eight calibration points based on microfossils of cyanobacteria and algae and geochemical evidence for methanogenesis and oxygenic photosynthesis (Supplementary Table 4). Many of these have been applied previously (e.g., refs. [55,89–95], but this is the first time they have been combined in a single molecular clock (Supplementary Fig. 21).

For each molecular clock, two independent chains were run for 43,650 to 156,348 cycles, and were considered converged when the relative difference of all parameters was > 0.3 and the effective sizes of all parameters were > 50. This was assessed using trace-comp implemented in phylobayes v4.1[84], discarding 10,914 to 49,182 cycles as burn-in. A suitable burn-in was chosen based on traces viewed in Tracer v.1.7.0[96].

### Identification of phosphorus-cycling genes

We searched in our molecular clock genome taxa, for homologs of genes involved in key pathways for the degradation, transport and production of phosphonates, phosphite and hypophosphite using HMMER3[68]. To do this, we used HMM profiles downloaded from the NCBI's prokaryotic genome annotation pipeline (Supplementary Table 5) and an e-value threshold of 0.1. Where these were unavailable (e.g., for proteins associated with phosphite and hypophosphite transport and oxidation), searches were conducted using the same e-value threshold and custom HMM profiles built from experimentally characterised proteins (Supplementary Table 6) that were aligned in MAFFT v. 7.4[97] with L-INS-I.

The resulting hits were retrieved and aligned with MAFFT v.7.4[97] using E-large-INS-1. Next, they were trimmed to remove positions comprising > 85% gaps with trimAl v1.2rev59[70]. This cut-off was chosen because less aggressive trimming has been found to better recover well-established relationships[98]. Maximum likelihood trees were then generated with IQTREE v.2.0.3[73] under substitution models chosen by ModelFinder[75] with 1000 ultrafast bootstraps[79].

To the resulting trees, we annotated bit scores from the HMM searches using the 'Branch score' and 'Propagate attribute' modules in TreeViewer v2.0.1[81]. In addition to assigning bit score values to the leaves of the tree, we assigned bit score values to internal nodes based on the average score of their descendants. Any leaves representing homologs which passed the bitscore thresholds of their respective TIGRFAMs were also highlighted, so that contours could be drawn delimiting clades with the same degree of similarity as the query sequences. Leaves belonging to a monophyletic clade containing the query sequences and any others which passed the respective bit score thresholds were retained for further analyses, whilst the rest were discarded (Supplementary Figs. 22–35). Similar filtering methods have been employed to identify light-harvesting genes[91] and arsenic resistance genes[62] in prokaryotes.

## Phylogenetic analyses of phosphorus-cycling genes

After filtering, the remaining amino acid sequences of each protein family were aligned with MAFFT v. 7.4[97] and trimmed to remove positions comprising > 70% gaps with trimAl[70]. Bayesian phylogenetic trees were reconstructed using MrBayes v. 3.2.7a[99] with a mixed amino acid model prior, invariant sites and gamma-distributed site rates. Chains were considered converged when the average standard deviation of split frequencies was < 0.01, the potential scale reduction factor was between 1.00 and 1.02, and the ESS scores of all parameters were > 200 after the first 25% of iterations were discarded as burn-in. These were calculated with Mr Bayes and Tracer v1.7[96]. Bayesian phylogenetic trees of the two phosphate-transporting enzymes, PstS and PNaS, did not converge within three months, so maximum likelihood methodology was implemented instead, using IQ-TREE v2.0.3[73] with 1,000 ultrafast bootstraps[79] and a substitution model chosen by ModelFinder[75]. To ensure all complex mixture models were included, the following flags were used: -mrate E, I, G, I + G, R and -madd C10, C20, C30, C40, C50, C60, EX2, EX3, EHO, UL2, UL3, EX_EHO, LG4M, LG4X, CF4. Between 100 and 700 iterations were performed before the ultrafast bootstraps converged. A spot check on phosphonatase revealed that similar results were found regardless of whether MrBayes or IQ-TREE were used to reconstruct the evolutionary trees (Supplementary Fig. 36).

## Reconciliation of gene trees with the molecular clock

To estimate the timing for duplication, transfer, loss, and speciation events for each gene, each gene tree was reconciled with our time-calibrated species tree using ecceTERA v. 1.2.5[44]. Default settings were applied (event costs: HGT = 3, duplication = 2, loss = 1, speciation = 0; transfer to the dead allowed so HGTs can occur with unsampled lineages) with amalgamate = true to find the most parsimonious reconciled gene tree from a set of gene trees representing uncertainty in their topology[100]. These gene tree sets were obtained from the.t files of the Mr Bayes runs using a custom script which discarded the first 25 % as burn-in. From the output of these gene-tree-species-tree reconciliations (specifically the symmetric median reconciliation), we collected the timing of all speciations, transfers, duplications, and losses of each gene by editing Python scripts that were first designed and applied to time the evolution of sulphur-cycling enzymes[55]. These were applied with Python v. 2.7.5. Each horizontal gene transfer includes two separate events which are counted and output by ecceTERA; a departure of the gene from one lineage and a reception into another lineage. To prevent double-counting and over-inflating the number of transfer events, only reception events are presented.

## Phosphate concentration

To investigate whether organisms with low-affinity phosphate transporters and CP-lyases are found in environments with high or low phosphate concentrations, we examined the Ocean Microbial Reference Catalogue v2 (OM-RGC.v2) from the *Tara* Oceans Project[48,49]. The OM-RGC.v2 included relative abundances of all clusters of orthologous genes (COG) in 139 *Tara* Oceans metagenomic samples. We associated each gene of interest with a unique COG accession number using the COG search tool (https://www.ncbi.nlm.nih.gov/research/cog/). To quantify how gene abundance varies with phosphate concentration, regression models of the abundance of each COG were fitted to the raw phosphate concentration using the 'lm' function of the 'stats' package in R v.4.2.2. All analyses include 129 degrees of freedom.

## Reporting summary

Further information on research design is available in the Nature Portfolio Reporting Summary linked to this article.

## Data availability

The alignment data, protein phylogenies, molecular clocks and species tree generated in this study are available in the open science framework repository, https://osf.io/vt5rw/?view_only=b13a53f4d87c44d1a82a18b176523c5b. Source data for Figs. 3 and 4 and Supplementary Figs. 1–16 are also available as a source data file. Accession numbers of all genomes analysed in this study are provided in Supplementary Data 1 and accession numbers of HMM profiles and query sequences are provided in supplementary tables 5 and 6. Source data are provided with this paper.

## Code availability

Custom scripts designed and applied in this project to obtain the last 75% of trees outputted from MrBayes and to parse the results of gene-tree-species-tree-reconciliations performed in ecceTERA are available in the open science framework repository, https://osf.io/vt5rw/?view_only=b13a53f4d87c44d1a82a18b176523c5b.

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

## Acknowledgements

Thanks are offered to several influential people who helped make this manuscript possible. These include Carlota Boch Varela, who conducted some of the gene-tree-species-tree-reconciliations with chronograms made using different calibration points and event costs. We also thank Aya Klos and Tony Ni for their helpful comments on the manuscript and Aya Klos for her advice on R scripting for figures. We also thank Dr. Celine Scornavacca for advice on ecceTERA, Dr. Abu Baidya and Dr. Matt Pasek for helpful discussions and Dr. Isabelle Bi for advice on statistical analyses. Funding for this work came from a NERC Frontiers grant (NE/V010824/1) awarded to E.E.S. R.A. performed this work as part of the Virtual Planetary Laboratory Team, a member of the NASA Nexus for Exoplanet System Science, funded via NASA Astrobiology Program Grant No. 80NSSC18K0829.

## Author contributions

E.E.S. R.E.A. and J.S.B conceived the research project; J.S.B and R.E.A. designed the phylogenetic and molecular clock analyses; J.S.B.

performed the analyses; J.Z. helped create Python scripts for parsing reconciliations; J.Z. and J.S.B. found *Tara* data and performed regression analyses; J.S.B, E.E.S. and R.E.A. interpreted the data; all authors wrote and reviewed the manuscript.

## Competing interests

The authors declare no competing interests.
