## [Peer Review File · Nature Communications]

Timing the Evolution of Phosphorus-Cycling Enzymes Through Geological Time using PhylogenomicsREVIEWER COMMENTS

Reviewer #1 (Remarks to the Author):

In this manuscript, the authors present a molecular clock-based analysis of the evolution of phosphorous-metabolizing genes in Bacteria and Archaea. The authors perform phylogenetic and molecular clock analyses of gene and species trees to determine the evolutionary history of these genes and relate their appearance in genomes to predicted amounts of different phosphorous species in the ocean of early Earth. Overall, the topic is of interest to researchers trying to reconstruct the conditions of early Earth and early life's evolution.

The manuscript is well written with an apparently straightforward series of analyses. However, this simplicity comes at a cost as important details of the phylogenetic and molecular clock analyses are missing. My major concerns are:

1. Phylogenetics: the authors hand-pick ~850 species to reconstruct the phylogeny that they use for all their inferences. While I appreciate the difficulty of estimating large tree of life phylogenies, taxon sampling has been shown to affect the topology of trees, and even more so in the case of prokaryotes where HGT and other factors have such a strong influence on the topology. The cursory overview of the topology estimated by the authors (lines 86-90) does not even scratch the surface of the complexity of prokaryote phylogenies and does not allow to truly evaluate the robustness of this reconstruction. As an example, the authors use the clustering of eukaryotes with Asgard archaea as support to the similarity of their phylogeny to previous ones but this support has a bootstrap of 21, which is very low, even for prokaryotic standards. I was not able to see anywhere, a full list of the species selected, an explanation for their selection, and a detailed phylogeny (either a newick file or a figure that can be zoomed in to look at the details of the whole tree of life).

a. A correlated point is the choice of using 16 ribosomal proteins for the alignment. Also in this case, I empathize with the complexity of the analysis the authors are trying to perform but the use of only a few genes has been shown to affect topology and divergence times. At the very least, a discussion of such potential issues should be present.

2. Timetree: similarly to the previous point, the authors describe divergence times in a limited way, focusing only on few major time periods (not even relaying specific nodes with their estimated times). The calibrations selected are primarily constrained within the cyanobacteria clade, which is not surprising given they are one of the few groups with information. However, also in this case, there is a large body of literature that discusses the effect of calibration points and their boundaries on estimated times. Yet, there is no mention of this in the manuscript and, apparently, no evaluation of the robustness of their results based on different choices.

3. Gene/species trees: a fundamental aspect of this analysis is the reconciliation of gene and species trees that has allowed the authors to identify HGT, duplications, and losses and, therefore, estimate the evolution of these genes. I was quite surprised to see that all the information provided effectively mounted to default settings in ecceTERA. The reconciliation of deep gene and species evolutionary history is complex and, surely, affected by the settings chosen in a software, at the very minimum.

In summary, I appreciate the effort of the authors and find the topic of great interest. However, I find the description of the study too superficial to truly reflect the complexity of prokaryote evolution. The apparent reliance of the authors on a simplistic methodology for phylogenetic, molecular clock, and gene/species tree reconstruction raises concerns on the validity of the results obtained, as they heavily depend on these three aspects of the analysis.

Reviewer #2 (Remarks to the Author):

Boden et al., present a fascinating new study on the history of phosphorus cycling. Their exploration is through molecular clocks where they date the origins of different enzymes and map this history onto chemical and modern oceanographic observations to estimate a qualitative history of phosphorus concentrations.

This paper is well organized, written, and presented, however I do have some reservations and suggestions which I detail below.

First, environment is dealt with in a very loose way throughout this manuscript. It is not made clear enough to the reader that local environments could be driving the changes observed without any influence from global seawater. While in parts of the text the authors are careful to note that they cannot identify what environments their high-P or low-P inferences suggest, in other parts of the manuscript they leave the reader with the impression that they are constraining marine phosphate levels. I don't think that being careful with language here takes away from the manuscript as I find the authors insights important regardless of the local-global nature. Therefore I think stating from the outset the limitations of this approach are important.

Second the authors utilize the Tara Oceans data set to explore relationships between their inferences and possible environment. However, the Tara Oceans project only looks at the surface ocean, I think it would be a good idea for the authors to explore other resources available such as the Ocean Protein Portal (<https://www.oceanproteinportal.org/>) to see if their results display interesting trends at deeper depths.

Third, I think it is important that the authors explore the role of cell population in their molecular clock analysis. For example, recent works have drawn attention to this point that some uncertainty of molecular clocks may be masked by the assumption of a fixed population size (Crockford et al., 2023). I think this is an important point for the authors to explore.

Fourth, I think the inference regarding BIF deposition is interesting but it is very speculative. Again, I think the paper provides important insights into the history of the P cycle in terms of ordering of events, however, I think attempting to provide constraints on marine phosphorus concentrations or linking findings to the BIF record which itself is mired in controversy, detracts from this work.

L35 Refs: Laakso and Schrag 2014, 2017, 2019; Derry 2015

L160 – what is meant by outperform? As the results of such explorations are often untestable with the sedimentary record.

L228 – but not 1000 times higher than all environments.

Reviewer #3 (Remarks to the Author):

In "Timing the Evolution of Phosphorus-Cycling Enzymes Through Geological Time" Boden and co-authors present new estimates on the timing of the origin of specific phosphorus-using enzymes, and then tie these findings to a history of phosphate availability on the early earth. The paper uses phylogenetic analysis to identify the history of microbial genes that use phosphorus in various speciations. They conclude that phosphate has been used for the longest time, followed by phosphonates, then by phosphite and hypophosphite. The findings are novel and of interest to the early-life community, and to those who study biogeochemical element cycling, especially around 2.5 billion years ago.

Admittedly I am not wholly qualified to comment on the use of genomic timelines for estimating ages. I'd defer to other reviewers' expertise to evaluate the methods in the paper. My comments below focus mostly on the conclusions and discussion. I first have some comments for 'discussion', which may allow for some expansion of the manuscript, and I highlight a major concern (which may just be a typo) that requires further explanation. I'm happy to recommend publication after some of the discussion and the major concern is addressed.

Discussion

The paper presents an interesting finding, which seems to disprove the 'long half-life' for phosphite (e.g., Herschy et al. 2018). In other words, the likelihood that phosphite as a major phosphorus species throughout the Archean and into the Proterozoic seems to be disproved. That's an interesting finding but may be reasonable given that the 'long half-life' was based on a solution of

phosphite sealed under air that was kept on a bench for 5 years that showed ~0.1% oxidation after that time. Other experiments that argue for a long half-life are based on the slow oxidation of phosphite salts (e.g., CaHPO_3) that require high temperatures (~500°C) in order to commence oxidation.

With the above noted, shorter half-lives of phosphite may be feasible, for example Gull et al. (Life, 2023) demonstrate that solutions of phosphite when heated to ~80°C in the presence of urea $[(\text{NH}_2)_2\text{CO}]$ produce phosphate and polyphosphates, from the oxidation of phosphite. That suggests that there may be other routes to oxidizing phosphite that were not considered in these 'long half-life' estimates.

In general, the oxidative half-life of hypophosphite appears to be much shorter than phosphite, and therefore the accumulation of hypophosphite in the geologic environment would be uncommon as water is sufficient to oxidize it.

The paper is pretty cautious about going further back in time than the mesoArchean (for example, at the origin of life, which makes sense given the tools at hand). Do the authors have any idea on the timeline required to lose a gene if, for example, a component 'disappears' from the environment? If phosphite were to diminish to nM or sub-nM concentrations, would the genes to use phosphite still exist if such low-phosphite conditions were present on the early earth for ~100 million years? 500 million years? If such genes would not persist then this may be worth including in the discussion.

The near-contemporaneous origin for phosphonate production and phosphite utilization suggests the two may be linked. Indeed, the breakdown of phosphonopyruvate should be sufficient to produce either phosphate or phosphite (both are exergonic reactions). Therefore, once anabolism of phosphonates began in a widespread fashion, phosphite was likely a side-product of this process. This may explain why the two appear somewhat contemporaneous.

Major

The description of phosphonate utilization (catabolism and production) at lines 162-182 does not seem to line up with Figure 3. Namely, "Microorganisms began catabolising phosphonates with the evolution of phnX in the Mesoarchean" and "Biological methods of producing these phosphonates began to emerge in the late Neoarchean,". Figure 3, if the labeling is correct, has pink for production beginning in the MesoArchean, and green for catabolism in the neoArchean. Either the figure caption or the text is in error.

Given the textual description, it is likely the colors are mixed up in Figure 3 (or my reading is incorrect). In this case, phosphonate catabolism predates phosphonate anabolism. This still strikes me as unusual. There are sparingly few natural abiotic sources of phosphonates in the environment. The only place they have been found has been in the Murchison meteorite, where they are about 0.1% of the total P content of the meteorite (the remainder is phosphates and phosphides), and an influx of carbonaceous material bearing phosphonic acids seems unlikely. How would catabolism have originated prior to having a source of phosphonates? If production is before catabolism, that would make more sense, but is not the argument presented in the paper.

Similar to the above, the paper does link their findings to the estimated abundance of phosphate in the geological environment, and those findings seem consistent. However, for phosphite, phosphonate, and hypophosphite, whether these are presumed to be in the environment or not is less clear. Are the authors arguing that these are trace components in the environment that have always been present, and then when phosphate decreased to such a level (in some environments) that there became an evolutionary pressure to start uptake of these molecules? Or is the manuscript arguing that these molecules began to become more prevalent in the environment at the timing of the evolutionary uptake for these (for example, through biological production)? It seems maybe the former for phosphite and the latter for phosphonate? This could be made clearer. It is easier to envision an abiotic source of phosphite than of phosphonates.

Line 287 "In the absence of iron oxidation, all iron would have been in the Fe(II) state, which renders phosphate relatively more soluble, possibly allowing phosphate concentrations of hundreds to thousands of μM ". I think this is probably too strong for the conclusion. For example, Fe^{2+} with phosphate can form vivianite, which the Brady paper does discuss, but dismisses at the higher pH (~ 8). Further, one of the most common Fe-minerals is magnetite which is $\text{Fe}_2+(\text{Fe}^{3+})_2\text{O}_4$. Magnetite would certainly have been present at this time.

Minor changes

---Check the spelling of "phosphorus" at 115.

---"depleted" at 282

---Figure 5 is somewhat confusing. The arrows are unclear. Perhaps putting phosphate (genomic estimates) in black but dashed lines may help, as the "geochemical estimates" are also for phosphate.

RESPONSE TO REVIEWERS' COMMENTS

Text is colour-coded to represent reviewer comments (black) and the author's responses (blue).

Reviewer #1 (Remarks to the Author):

In this manuscript, the authors present a molecular clock-based analysis of the evolution of phosphorous-metabolizing genes in Bacteria and Archaea. The authors perform phylogenetic and molecular clock analyses of gene and species trees to determine the evolutionary history of these genes and relate their appearance in genomes to predicted amounts of different phosphorous species in the ocean of early Earth. Overall, the topic is of interest to researchers trying to reconstruct the conditions of early Earth and early life's evolution.

The manuscript is well written with an apparently straightforward series of analyses. However, this simplicity comes at a cost as important details of the phylogenetic and molecular clock analyses are missing. My major concerns are:

1. Phylogenetics: the authors hand-pick ~850 species to reconstruct the phylogeny that they use for all their inferences. While I appreciate the difficulty of estimating large tree of life phylogenies, taxon sampling has been shown to affect the topology of trees, and even more so in the case of prokaryotes where HGT and other factors have such a strong influence on the topology. The cursory overview of the topology estimated by the authors (lines 86-90) does not even scratch the surface of the complexity of prokaryote phylogenies and does not allow to truly evaluate the robustness of this reconstruction. As an example, the authors use the clustering of eukaryotes with Asgard archaea as support to the similarity of their phylogeny to previous ones but this support has a bootstrap of 21, which is very low, even for prokaryotic standards. I was not able to see anywhere, a full list of the species selected, an explanation for their selection, and a detailed phylogeny (either a newick file or a figure that can be zoomed in to look at the details of the whole tree of life).

A full list of the taxa and a newick file of our species tree has been added to our open science framework repository here:

https://osf.io/vt5rw/?view_only=b13a53f4d87c44d1a82a18b176523c5b. They are entitled 'TaxaList.csv' and 'Q2_merged_constraint3.treefile'.

We have also expanded on the relationships between phyla on lines 87 to 92 to help readers evaluate topology.

An explanation of why we chose the 865 taxa was presented in the methods section under 'Species Tree Reconstruction'. We were attempting to sample across the diversity of life (as currently understood), but are aware that other methods of sample selection exist (e.g. the prototype selection algorithm applied in (Zhu et al., 2019) and the quantile-based equations of (Martinez-Gutierrez and Aylward, 2021)). One genome per order with the addition of 12 cyanobacteria and plastids for calibration purposes were chosen to balance broad and diverse sampling with computational restrictions.

- a. A correlated point is the choice of using 16 ribosomal proteins for the alignment. Also in this case, I empathize with the complexity of the analysis the authors are

trying to perform but the use of only a few genes has been shown to affect topology and divergence times. At the very least, a discussion of such potential issues should be present.

A table has been compiled summarising the number of proteins used in other tree of life studies (Supplementary Information, Table 2 below). Although a couple have used in the region of 300-400 proteins, most use 14-30 to reconstruct the tree of life and estimate divergence times. The exception is Zhu et al. (2019), where a dataset of 381 proteins were used to reconstruct the tree of life, but these had to be down-sampled to an alignment of 5,000 amino acid sites (equivalent to ~16 proteins if prokaryotic genes span 924 nucleotides (Xu et al., 2006)) to run the molecular clock. Some extra sentences have been added to the methods for ‘Species Tree Reconstruction’ to explain that other methods of reconstructing phylogenies are available.

Table 2: The ribosomal proteins used here to reconstruct the tree of life and molecular clock have been utilised in several independent studies for a similar purpose. A complete representation of papers reporting on the topology of the tree of life, including Bacteria and Archaea is included alongside the presence or absence of the ribosomal proteins used in this analyses in their multi-gene datasets. The total number of proteins used in each dataset is listed on the right alongside a column detailing whether they also ran a molecular clock using the same proteins.

Paper	Ribosomal Protein															Number of Proteins	Type of Tree	Molecular Clock	
	Rpl2	Rpl3	Rpl4	Rpl5	Rpl6	Rpl14	Rpl15e	Rpl16	Rpl18	Rpl22	Rpl24	Rps3	Rps8	Rps10	Rps17				Rps19
Sunagawa et al, 2013	Y	Y	Y	Y	Y	Y	Y	Y	Y	Y	n	Y	Y	n	Y	Y	40	All	n/a
Segata et al, 2013	?	?	?	?	?	?	?	?	?	?	?	?	?	?	?	?	400	All	n/a
Hug et al, 2016	Y	Y	Y	Y	Y	Y	Y	Y	Y	Y	Y	Y	Y	Y	Y	Y	16	All	n/a
Castelle and Banfield et al, 2018	Y	Y	Y	Y	Y	Y	Y	n	Y	Y	Y	Y	Y	n	Y	Y	14	All	n/a
Betts et al, 2018	Y	n	n	n	n	n	n	n	n	n	n	Y	n	n	n	n	29	All	same
Zhu et al, 2019	Y	n	n	Y	n	n	n	n	n	n	n	Y	n	n	n	n	381	All	same
Chen et al, 2020	Y	Y	Y	Y	Y	Y	Y	Y	Y	Y	Y	Y	Y	Y	Y	Y	16	All	same
Williams et al, 2020	n	n	n	n	n	n	n	n	n	n	n	Y	n	n	n	n	21	All	n/a
Martinez-Gutierrez et al, 2021	Y	Y	Y	Y	Y	Y	Y	Y	Y	Y	Y	Y	n	Y	Y	Y	30	All	n/a
Moody et al, 2022	Y	Y	n	Y	n	Y	n	n	n	Y	n	Y	Y	Y	n	n	27	All	same
Martinez-Gutierrez et al, 2023	Y	Y	Y	Y	Y	Y	Y	Y	Y	Y	n	Y	Y	n	Y	Y	30	All	same

2. Timetree: similarly to the previous point, the authors describe divergence times in a limited way, focusing only on few major time periods (not even relaying specific nodes with their estimated times). The calibrations selected are primarily constrained within the cyanobacteria clade, which is not surprising given they are one of the few groups with information. However, also in this case, there is a large body of literature that discusses the effect of calibration points and their boundaries on estimated times. Yet, there is no mention of this in the manuscript and, apparently, no evaluation of the robustness of their results based on different choices.

We understand your concern. Calibration points are key determining factors for any molecular clock study, so it's important to choose values which are accurate. We thought carefully about which calibration points to apply, and chose fossil and geochemical evidence that has been verified and published. For example, our calibrations on the origin of oxygenic photosynthesis, akinetes and cyanobacterial symbionts have been applied in multiple papers by Sanchez-Baracaldo et al (e.g. (Sanchez-Baracaldo, 2015, Sanchez-Baracaldo et al., 2005, Sanchez-Baracaldo et al., 2017, Sanchez-Baracaldo et al., 2014) and our calibrations for the origin of red algae, eukaryotes and methanogenesis were applied in (Mateos et al., 2023, Parsons et al., 2021). Previous work has tested different age constraints for these latter three calibrations based on broadly accepted values in the literature (so-called "conservative" calibration points) and the earliest available evidence (so-called "liberal")(Parsons et al.,

2021, Mateos et al., 2023). The liberal points yielded unrealistic ages for the origin of LUCA, before the origin of planet Earth, so these were discarded and the conservative points were applied here.

Recent geochemical evidence on the origin of methanogenesis suggests that it may have evolved earlier than the >2.7 Ga calibration we initially applied. To investigate this, we moved this calibration earlier to >3.46 Ga to reflect geological evidence of methanogenesis 3.42 Ga from the Barberton Greenstone Belt (Cavalazzi et al., 2021) and 3.46 Ga in the Pilbara Craton (Ueno et al., 2006) as well as phylogenetic evidence derived from horizontal gene transfers which indicate methanogenic Euryarchaeota 3.51 Ga (Wolfe and Fournier, 2018). We ran this with the lognormal model and found that it increases the age of crown Archaea from 3.26 Ga (confidence intervals span 3.45 to 3.08 Ga) to 3.74 Ga (confidence intervals span 3.83 to 3.66 Ga). Few phosphorus-cycling genes are found within Archaea (Figure 2), and the older calibration has little impact on crown bacteria, so when we reconcile our protein trees with this new clock, there is little impact on the timing of evolution of phosphorus-cycling enzymes (Figure below).

Figure: Calibrating methanogenesis to evolve earlier in Earth history has little impact on the estimated timing of the evolution and spread of phosphorus-cycling enzymes. One of seven calibration points is altered to force methanogenesis to evolve in the MRCA of TACK and Euryarchaeota > 3.46 Ga instead of > 2.7 Ga to reflect the findings of (Ueno et al., 2006, Wolfe and Fournier, 2018, Cavalazzi et al., 2021). These reconciliations were calculated using results of molecular clocks made with the lognormal clock model. Mya, million years ago; GOE, great oxygenation event.

3. Gene/species trees: a fundamental aspect of this analysis is the reconciliation of gene and species trees that has allowed the authors to identify HGT, duplications, and losses and, therefore, estimate the evolution of these genes. I was quite surprised to see that all the information provided effectively mounted to default settings in ecceTERA. The reconciliation of deep gene and species evolutionary history is complex and, surely, affected by the settings chosen in a software, at the very minimum.

We appreciate that default settings can reflect a lack of knowledge from the authors, but this was not the case here. We amalgamated our gene trees to account for uncertainty in their topology and rooting and transfer to the dead was allowed to mitigate the impact of reduced taxon sampling (many orders, families, genera and species could have received genes via lateral gene transfer despite not being represented in the time-calibrated tree of life). While these are well-founded methodological decisions, the relative cost of transfer vs. duplication vs. loss vs. speciation can be debated, so in response to these concerns we have modified these parameters to assess how they impact our conclusions.

The default costs for gene transfers, duplications, losses and speciations have been found to minimise genome size variation between parent and daughter lineages (David and Alm, 2011). Whether genome size would have remained constant between parent and daughter lineages on timescales of the order of hundreds of millions of years can be debated. Some findings suggest that genome size distributions are strongly linked to evolutionary history (Martinez-Gutierrez and Aylward, 2022) implying little change across lineages. In contrast, others have found that terrestrial prokaryotes have larger genomes than aquatic and host-associated prokaryotes, implying that related organisms from different habitats would have different genome sizes (Rodríguez-Gijón et al., 2021).

If genome size does vary between parent and daughter lineages that span millions of years, and David and Alm's findings are valid, then HGT costs would have been higher (David and Alm, 2011). Our results suggest that when HGT is more expensive (costing 4 or 6 units instead of 3), phosphorus-cycling enzymes evolve in the same general order with mechanisms of phosphate import emerging in the Paleoarchean, followed by mechanisms of phosphonate catabolism in the Mesoarchean, followed by phosphite utilisation and phosphonate production in the Neoarchean and during the great oxygenation event (see figure below). The main differences lie in the timing of evolution of the high-affinity phosphate uptake mechanism, *pstS*. When HGT is more costly, *pstS* is estimated to have evolved around the same time as *pnas*, not later as default costs imply. Therefore, microbes may have lived in environments with low concentrations of phosphorus for longer, since the Paleoarchean. Similarly, the timing of emergence of environments with very low phosphate concentrations could have been earlier because *phnJ* and *phnM* are estimated to evolve

before the GOE, as opposed to after it or towards the end. Linking all scenarios is a lag spanning a few hundred million years between the emergence and spread of *pstS* (indicating low phosphate concentrations) and *phnJ* and *phnM* (indicating very low phosphate concentrations).

Figure: Effect of higher HGT costs on the estimated timing of the evolution and spread of phosphorus-cycling enzymes. HGT cost is varied from the default of 3 (a) to 4 (b) and 6 (c) whilst retaining the default costs of gene duplication, loss and speciation. These

reconciliations were calculated using results of molecular clocks made with the CIR clock model. Mya, million years ago; GOE, great oxygenation event.

If HGT was cheaper, costing 2 instead of 3 units whilst all other events remain the same, the results remain broadly similar to those with default parameters, although we cannot model the evolution of *pstS* because the number of optimal reconciliation is too large for ecceTERA to compute (figure below). We have modified the discussion and included these in the supplementary information to demonstrate the uncertainty.

Figure: Effect of lower HGT costs on the estimated timing of the evolution and spread of phosphorus-cycling enzymes. HGT cost is varied from the default of 3 (a) down to 2 (b) whilst retaining the default costs of gene duplication, loss and speciation. These reconciliations were calculated using results of molecular clocks made with the CIR clock model. Mya, million years ago; GOE, great oxygenation event.

The manuscript has been edited to include the results of these runs.

In summary, I appreciate the effort of the authors and find the topic of great interest. However, I find the description of the study too superficial to truly reflect the complexity of prokaryote evolution. The apparent reliance of the authors on a simplistic methodology for phylogenetic, molecular clock, and gene/species tree reconstruction raises concerns on the validity of the results obtained, as they heavily depend on these three aspects of the analysis.

Reviewer #2 (Remarks to the Author):

Boden et al., present a fascinating new study on the history of phosphorus cycling. Their exploration is through molecular clocks where they date the origins of different enzymes and map this history onto chemical and modern oceanographic observations to estimate a qualitative history of phosphorus concentrations.

This paper is well organized, written, and presented, however I do have some reservations and suggestions which I detail below.

First, environment is dealt with in a very loose way throughout this manuscript. It is not made clear enough to the reader that local environments could be driving the changes observed without any influence from global seawater. While in parts of the text the authors are careful to note that they cannot identify what environments their high-P or low-P inferences suggest, in other parts of the manuscript they leave the reader with the impression that they are constraining marine phosphate levels. I don't think that being careful with language here takes away from the manuscript as I find the authors insights important regardless of the local-global nature. Therefore I think stating from the outset the limitations of this approach are important.

Fair enough, we've modified our wording to help clarify this in the discussion.

Second the authors utilize the Tara Oceans data set to explore relationships between their inferences and possible environment. However, the Tara Oceans project only looks at the surface ocean, I think it would be a good idea for the authors to explore other resources available such as the Ocean Protein Portal (<https://www.oceanproteinportal.org/>) to see if their results display interesting trends at deeper depths.

Cool resource, thanks! The Ocean Gene Atlas (<https://tara-oceans.mio.osupytheas.fr/ocean-gene-atlas/>) creates some nice graphs of gene abundance with depth, but seems to be limited to a smaller dataset. There are no records of any of the phosphonate cycling enzymes (namely PhnJ, PhnM, PhnZ, PhnX, PhnA, PhnW, Ppd or PepM), or the phosphite cycling enzymes (PtxB and PtxD) or HtxB or PNaS in any of their records from the surface ocean to 6,000m depth, but there are records of PstS, PNaS and HtxA. HtxA is present in a single *Marinobacter* strain between 50 and 800m depth, whereas PstS is present in 4 strains beneath 200m: *Prochlorococcus*, *Limnobacter*, *Alcanivorax* and *Teredinibacter*. PNaS is in *Desulfococcus* up to 1,250m depth. These differences are surprising given the detection of some of these genes in the Tara dataset, so may come down to differences in sampling strategies and processing. Previous studies (e.g. (Lockwood et al., 2022)) have looked for correlations of phosphorus-cycling genes with phosphate concentrations at depth, and have found little correlation between phosphate concentration and the abundance of phosphorus-cycling enzymes at depth. This

could be because phosphate concentrations at depth are higher, so phosphonates are being used as a source of carbon or oxygen instead of phosphorus. Sentences have been added to the discussion to explain that the availability of other molecules could also influence the evolution of phosphonate catabolism genes.

Third, I think it is important that the **authors explore the role of cell population** in their molecular clock analysis. For example, recent works have drawn attention to this point that some uncertainty of molecular clocks may be masked by the assumption of a fixed population size (Crockford et al., 2023). I think this is an important point for the authors to explore.

Crockford et al (2023) make some good points about how the potential for evolution can change based on the size of the population. In a larger population, there are more replications in a given time-frame and environment due to the number of cells replicating. As a result, there is more potential for mutation, so if the number of cells on Earth has increased over geological time in-line with productivity (Crockford et al, 2023), the potential for mutation has also increased. This is partially accounted for in our molecular clocks because they are relaxed, meaning that mutation rates can vary over time. These rates vary between lineages, but as yet no method of accounting for variation within lineages exist. As a result, the only way to improve resolution through time is to include more genomes in the analyses. We were limited to including a few hundred genomes here because adding more genomes to improve resolution in rate variation prevents the molecular clocks from converging in a reasonable amount of time, even on high performance computing clusters. Therefore, we add a sentence to the discussion to point out this limitation:

“The inclusion of novel lineages, or new models of evolution to account for within-lineage variation in mutation rates may alter our conclusions”

Fourth, I think the inference regarding BIF deposition is interesting but it is very speculative. Again, I think the paper provides important insights into the history of the P cycle in terms of ordering of events, however, I think attempting to provide constraints on marine phosphorus concentrations or linking findings to the BIF record which itself is mired in controversy, detracts from this work.

We have reworded this point and emphasized that is just speculative. However, we prefer to keep this suggestion in the manuscript, because it may offer a potential explanation for the data.

L35 Refs: Laakso and Schrag 2014, 2017, 2019; Derry 2015

Great papers, thanks! Added.

L160 – what is meant by outperform? As the results of such explorations are often untestable with the sedimentary record.

CIR outperforms other clock models (namely LN and UGAM) because it produces better Bayes factors than UGAM (Lepage et al., 2007) and produces dates which are more consistent than LN and UGAM with relative dates of different lineages that have participated in the same horizontal gene transfers (Fournier et al., 2021). Bayes factors test how well a model of evolution fits the alignment used to generate a molecular clock, so both of these methods are independent of any geological records. I’ve added some extra words to clarify:

“we focus on estimates from the CIR clock model because it has been shown to outperform other clock models in terms of model fit⁴⁴ and consistency with relative dates of lineages participating in the same HGT events⁴⁵”

L228 – but not 1000 times higher than all environments.

Added the word 'marine' to clarify

Reviewer #3 (Remarks to the Author):

In “Timing the Evolution of Phosphorus-Cycling Enzymes Through Geological Time” Boden and co-authors present new estimates on the timing of the origin of specific phosphorus-using enzymes, and then tie these findings to a history of phosphate availability on the early earth. The paper uses phylogenetic analysis to identify the history of microbial genes that use phosphorus in various speciations. They conclude that phosphate has been used for the longest time, followed by phosphonates, then by phosphite and hypophosphite. The findings are novel and of interest to the early-life community, and to those who study biogeochemical element cycling, especially around 2.5 billion years ago.

Admittedly I am not wholly qualified to comment on the use of genomic timelines for estimating ages. I'd defer to other reviewers' expertise to evaluate the methods in the paper. My comments below focus mostly on the conclusions and discussion. I first have some comments for 'discussion', which may allow for some expansion of the manuscript, and I highlight a major concern (which may just be a typo) that requires further explanation. I'm happy to recommend publication after some of the discussion and the major concern is addressed.

Discussion

The paper presents an interesting finding, which seems to disprove the 'long half-life' for phosphite (e.g., Herschy et al. 2018). In other words, the likelihood that phosphite as a major phosphorus species throughout the Archean and into the Proterozoic seems to be disproved. That's an interesting finding but may be reasonable given that the 'long half-life' was based on a solution of phosphite sealed under air that was kept on a bench for 5 years that showed ~0.1% oxidation after that time. Other experiments that argue for a long half-life are based on the slow oxidation of phosphite salts (e.g., CaHPO_3) that require high temperatures (~500°C) in order to commence oxidation.

With the above noted, shorter half-lives of phosphite may be feasible, for example Gull et al. (Life, 2023) demonstrate that solutions of phosphite when heated to ~80°C in the presence of urea $[(\text{NH}_2)_2\text{CO}]$ produce phosphate and polyphosphates, from the oxidation of phosphite. That suggests that there may be other routes to oxidizing phosphite that were not considered in these 'long half-life' estimates.

In general, the oxidative half-life of hypophosphite appears to be much shorter than phosphite, and therefore the accumulation of hypophosphite in the geologic environment would be uncommon as water is sufficient to oxidize it.

The paper is pretty cautious about going further back in time than the mesoArchean (for example, at the origin of life, which makes sense given the tools at hand). Do the authors have any idea on the timeline required to lose a gene if, for example, a component 'disappears' from the environment? If phosphite were to diminish to nM or sub-nM concentrations, would the genes to use phosphite still exist if such low-phosphite conditions were present on the early earth for ~100 million years? 500

million years? If such genes would not persist then this may be worth including in the discussion.

How quickly would a gene disappear if the phosphorus-compound it acts upon disappears from the environment? It's difficult to know. Genes can appear and disappear from populations in a few months or years (Bendall et al., 2016, Anderson et al., 2022), but these populations are relatively small, spanning just one hydrothermal vent or lake. To understand how a gene could be lost from an entire lineage which probably has representatives in hundreds of populations in several different geographical areas, would require every place and population containing members of that lineage to experience the same. To drive this would require a global loss of the molecules these genes act on. That global loss would also have to persist because genes are often retained in populations if they are occasionally useful. If the global average dropped from micromolar to sub-nanomolar concentrations of phosphite, but a few environments (e.g. deep subsurface) retained higher concentrations and microbial populations, then the genes for phosphite import and oxidation would remain. However, if the global average dropped by the same amount, but no phosphite oases existed, the gene would be lost. To predict how quickly a gene would be lost from global biota requires an understanding of habitat heterogeneity, which we can't predict using the methods in this manuscript.

The near-contemporaneous origin for phosphonate production and phosphite utilization suggests the two may be linked. Indeed, the breakdown of phosphonopyruvate should be sufficient to produce either phosphate or phosphite (both are exergonic reactions). Therefore, once anabolism of phosphonates began in a widespread fashion, phosphite was likely a side-product of this process. This may explain why the two appear somewhat contemporaneous.

Major

The description of phosphonate utilization (catabolism and production) at lines 162-182 does not seem to line up with Figure 3. Namely, "Microorganisms began catabolising phosphonates with the evolution of phnX in the Mesoarchean" and "Biological methods of producing these phosphonates began to emerge in the late Neoproterozoic". Figure 3, if the labeling is correct, has pink for production beginning in the MesoArchean, and green for catabolism in the neoArchean. Either the figure caption or the text is in error.

The figure caption was incorrect and has now been amended

Given the textual description, it is likely the colors are mixed up in Figure 3 (or my reading is incorrect). In this case, phosphonate catabolism predates phosphonate anabolism. This still strikes me as unusual. There are sparingly few natural abiotic sources of phosphonates in the environment. The only place they have been found has been in the Murchison meteorite, where they are about 0.1% of the total P content of the meteorite (the remainder is phosphates and phosphides), and an influx of carbonaceous material bearing phosphonic acids seems unlikely. How would catabolism have originated prior to having a source of phosphonates? If production is before catabolism, that would make more sense, but is not the argument presented in the paper.

It is possible that older genes generating phosphonates were lost from the genomic record. We have no way of assessing this scenario. Genes that did not survive were likely not widespread across the tree of life at the time, but they may have existed in high abundance at a local scale, enabling the emergence of phosphonate catabolizing enzymes. Alternatively, it is possible that phosphonates

were generated abiotically, perhaps in metamorphic or hydrothermal settings in the presence of phosphate and organic matter. To our knowledge, this scenario has so far not been tested experimentally, but it may potentially offer an explanation. We have added a brief discussion to this effect to the manuscript:

“We note that biological phosphonate production appears to have emerged after phosphonate utilization in our data, which may indicate an extinct phosphonate-generating pathway in the early Archean that was not sufficiently widespread to be preserved in the genomic record. Alternatively, it is possible that minor phosphonates were generated abiotically in metamorphic or hydrothermal settings in the presence of phosphate and organic matter, although experimental evidence for this is so far lacking. In any case, our results unambiguously show the Paleoproterozoic radiation of biological phosphonate uptake pathways, and the mere fact that this signal is preserved in modern organisms speaks to the significance of this metabolic pathway at that time.”

Similar to the above, the paper does link their findings to the estimated abundance of phosphate in the geological environment, and those findings seem consistent. However, for phosphite, phosphonate, and hypophosphite, whether these are presumed to be in the environment or not is less clear. Are the authors arguing that these are trace components in the environment that have always been present, and then when phosphate decreased to such a level (in some environments) that there became an evolutionary pressure to start uptake of these molecules? Or is the manuscript arguing that these molecules began to become more prevalent in the environment at the timing of the evolutionary uptake for these (for example, through biological production)? It seems maybe the former for phosphite and the latter for phosphonate? This could be made clearer. It is easier to envision an abiotic source of phosphite than of phosphonates.

We can't differentiate between these two hypotheses based on genomic data alone. All we can do is look for correlations between the abundance of a compound and the abundance of a gene to see if the gene is associated with different concentrations of phosphorus compounds. Some extra sentences have been added to the discussion section entitled 'Reconstructing phosphate availability through time' to explain.

Line 287 "In the absence of iron oxidation, all iron would have been in the Fe(II) state, which renders phosphate relatively more soluble, possibly allowing phosphate concentrations of hundreds to thousands of μM ". I think this is probably too strong for the conclusion. For example, Fe^{2+} with phosphate can form vivianite, which the Brady paper does discuss, but dismisses at the higher pH (~ 8). Further, one of the most common Fe-minerals is magnetite which is $\text{Fe}_2+(\text{Fe}^{3+})_2\text{O}_4$. Magnetite would certainly have been present at this time.

We softened the wording of this section and stress that the BIF explanation for the declining phosphate trend is somewhat speculating at this early stage. However, the high solubility proposed by Brady et al. (2022) stands at $\text{pH} < 8$, which is relevant for the Archean (Isson et al., 2018, Krissansen-Totton et al., 2018, Halevy and Bachan, 2017). Yes, magnetite dates back to 3.8 Ga; however, the total volume of BIF increased in the Neoproterozoic and Paleoproterozoic, and so it is plausible that oxides were on a global scale a much less important phosphate sink before that time.

Minor changes

---Check the spelling of "phosphorus" at 115.

amended

---“depleted” at 282

amended

---Figure 5 is somewhat confusing. The arrows are unclear. Perhaps putting phosphate (genomic estimates) in black but dashed lines may help, as the “geochemical estimates” are also for phosphate.

we’ve edited Figure 5 to make it clear that genomic estimates of phosphate concentrations are based on different genes.

References

- ANDERSON, R. E., GRAHAM, E. D., HUBER, J. A. & TULLY, B. J. 2022. Microbial Populations Are Shaped by Dispersal and Recombination in a Low Biomass Subseafloor Habitat. *mBio*, 13, e00354-22.
- BENDALL, M. L., STEVENS, S. L. R., CHAN, L.-K., MALFATTI, S., SCHWIENTEK, P., TREMBLAY, J., SCHACKWITZ, W., MARTIN, J., PATI, A., BUSHNELL, B., FROULA, J., KANG, D., TRINGE, S. G., BERTILSSON, S., MORAN, M. A., SHADE, A., NEWTON, R. J., MCMAHON, K. D. & MALMSTROM, R. R. 2016. Genome-wide selective sweeps and gene-specific sweeps in natural bacterial populations. *The ISME Journal*, 10, 1589-1601.
- BRADY, M. P., TOSTEVIN, R. & TOSCA, N. J. 2022. Marine phosphate availability and the chemical origins of life on Earth. *Nature Communications*, 13, 5162.
- CAVALAZZI, B., LEMELLE, L., SIMIONOVICI, A., CADY, S. L., RUSSELL, M. J., BAILO, CANTERI, R., ENRICO, E., MANCEAU, A., MARIS, A., SALOME, M., THOMASSOT, E., BOUDEN, N., TUCOULOU, R. & HOFMANN, A. 2021. Cellular remains in a ~3.42-billion-year-old subseafloor hydrothermal environment. *Sci Adv*, 7.
- COLEMAN, G. A., DAVIN, A. A., MAHENDRARAJAH, T. A., SZANTHO, L. L., SPANG, A., HUGENHOLTZ, P., SZOLLOSI, G. J. & WILLIAMS, T. A. 2021. A rooted phylogeny resolves early bacterial evolution. *Science*, 372.
- DAVID, L. A. & ALM, E. J. 2011. Rapid evolutionary innovation during an Archaean genetic expansion. *Nature*, 469, 93-96.
- FOURNIER, G. P., MOORE, K. R., RANGEL, L. T., PAYETTE, J. G., MOMPER, L. & BOSAK, T. 2021. The Archean origin of oxygenic photosynthesis and extant cyanobacterial lineages. *Proc Biol Sci*, 288, 20210675.
- HALEVY, I. & BACHAN, A. 2017. The geologic history of seawater pH. *Science*, 355, 1069-1071.
- ISSON, T. T., LOVE, G. D., DUPONT, C. L., REINHARD, C. T., ZUMBERGE, A. J., ASAEL, D., GUEGUEN, B., MCCROW, J., GILL, B. C., OWENS, J., RAINBIRD, R. H., ROONEY, A. D., ZHAO, M. Y., STUEEKEN, E. E., KONHAUSER, K. O., JOHN, S. G., LYONS, T. W. & PLANAVSKY, N. J. 2018. Tracking the rise of eukaryotes to ecological dominance with zinc isotopes. *Geobiology*, 16, 341-352.
- KRISSANSEN-TOTTON, J., ARNEY, G. N. & CATLING, D. C. 2018. Constraining the climate and ocean pH of the early Earth with a geological carbon cycle model. *Proceedings of the National Academy of Sciences*, 115, 4105-4110.
- LEPAGE, T., BRYANT, D., PHILIPPE, H. & LARTILLOT, N. 2007. A general comparison of relaxed molecular clock models. *Mol Biol Evol*, 24, 2669-80.
- LOCKWOOD, S., GREENING, C., BALTAR, F. & MORALES, S. E. 2022. Global and seasonal variation of marine phosphonate metabolism. *The ISME Journal*, 16, 2198-2212.
- MARTINEZ-GUTIERREZ, C. A. & AYLWARD, F. O. 2021. Phylogenetic Signal, Congruence, and Uncertainty across Bacteria and Archaea. *Molecular Biology and Evolution*, 38, 5514-5527.

- MARTINEZ-GUTIERREZ, C. A. & AYLWARD, F. O. 2022. Genome size distributions in bacteria and archaea are strongly linked to evolutionary history at broad phylogenetic scales. *PLOS Genetics*, 18, e1010220.
- MATEOS, K., CHAPPELL, G., KLOS, A., LE, B., BODEN, J., STUEKEN, E. & ANDERSON, R. 2023. The evolution and spread of sulfur cycling enzymes reflect the redox state of the early Earth. *Sci Adv*, 9, eade4847.
- MOODY, E. R. R., MAHENDRARAJAH, T. A., DOMBROWSKI, N., CLARK, J. W., PETITJEAN, C., OFFRE, P., SZOLLOSI, G. J., SPANG, A. & WILLIAMS, T. A. 2022. An estimate of the deepest branches of the tree of life from ancient vertically evolving genes. *Elife*, 11.
- PARSONS, C., STUEKEN, E. E., ROSEN, C. J., MATEOS, K. & ANDERSON, R. E. 2021. Radiation of nitrogen-metabolizing enzymes across the tree of life tracks environmental transitions in Earth history. *Geobiology*, 19, 18-34.
- RODRÍGUEZ-GIJÓN, A., NUY, J. K., MEHRSHAD, M., BUCK, M., SCHULZ, F., WOYKE, T. & GARCIA, S. L. 2021. A Genomic Perspective Across Earth's Microbiomes Reveals That Genome Size in Archaea and Bacteria Is Linked to Ecosystem Type and Trophic Strategy. *Front Microbiol*, 12, 761869.
- SANCHEZ-BARACALDO, P. 2015. Origin of marine planktonic Cyanobacteria. *Scientific Reports*, 5, 17418.
- SANCHEZ-BARACALDO, P., HAYES, P. K. & BLANK, C. E. 2005. Morphological and habitat evolution in the Cyanobacteria using a compartmentalization approach. *Geobiology*, 3, 145-165.
- SANCHEZ-BARACALDO, P., RAVEN, J. A., PISANI, D. & KNOLL, A. H. 2017. Early photosynthetic eukaryotes inhabited low-salinity habitats. *Proceedings of the National Academy of Sciences of the United States of America*, 114, E7737-E7745.
- SANCHEZ-BARACALDO, P., RIDGWELL, A. & RAVEN, J. A. 2014. A Neoproterozoic transition in the marine nitrogen cycle. *Current Biology*, 24, 652-657.
- UENO, Y., YAMADA, K., YOSHIDA, N., MARUYAMA, S. & ISOZAKI, Y. 2006. Evidence from fluid inclusions for microbial methanogenesis in the early Archaean era. *Nature*, 440, 516-519.
- WOLFE, J. M. & FOURNIER, G. P. 2018. Horizontal gene transfer constrains the timing of methanogen evolution. *Nat Ecol Evol*, 2, 897-903.
- XU, L., CHEN, H., HU, X., ZHANG, R., ZHANG, Z. & LUO, Z. W. 2006. Average gene length is highly conserved in prokaryotes and eukaryotes and diverges only between the two kingdoms. *Mol Biol Evol*, 23, 1107-8.
- ZHU, Q., MAI, U., PFEIFFER, W., JANSSEN, S., ASNICAR, F., SANDERS, J. G., BELDA-FERRE, P., AL-GHALITH, G. A., KOPYLOVA, E., MCDONALD, D., KOSCIOLEK, T., YIN, J. B., HUANG, S., SALAM, N., JIAO, J.-Y., WU, Z., XU, Z. Z., CANTRELL, K., YANG, Y., SAYYARI, E., RABIEE, M., MORTON, J. T., PODELL, S., KNIGHTS, D., LI, W.-J., HUTTENHOWER, C., SEGATA, N., SMARR, L., MIRARAB, S. & KNIGHT, R. 2019. Phylogenomics of 10,575 genomes reveals evolutionary proximity between domains Bacteria and Archaea. *Nature Communications*, 10, 5477.

REVIEWERS' COMMENTS

Reviewer #1 (Remarks to the Author):

I would like to thank the authors for the amount of work they added in response to the reviewers' comments. I believe these changes have significantly improved the manuscript and I am happy to recommend it for publication.

Reviewer #2 (Remarks to the Author):

I feel the authors have addressed my previous concerns and I am supportive of this MS for publication.

Reviewer #3 (Remarks to the Author):

The authors have addressed my points and relabeled the figure that was confusing me. I think the assumptions and findings have been better clarified. I'm happy with this version, but do note that the other reviewers had more comments that are more method-based than mine, and recommend careful consideration of those.